# Ligand-promoted cobalt-catalyzed radical hydroamination of alkenes

Xuzhong Shen[1], Xu Chen[1], Jieping Chen[1], Yufeng Sun[1], Zhaoyang Cheng[1] & Zhan Lu [1]*

Highly regio- and enantioselective intermolecular hydroamination of alkenes is a challenging process potentially leading to valuable chiral amines. Hydroamination of alkenes via metal-catalyzed hydrogen atom transfer (HAT) with good regioselectivity and functional group tolerance has been reported, however, high enantioselectivity has not been achieved due to the lack of suitable ligands. Here we report a ligand-promoted cobalt-catalyzed Markovnikov-type selective radical hydroamination of alkenes with diazo compounds. This operationally simple protocol uses unsymmetric *NNN*-tridentate (UNT) ligand, readily available alkenes and hydrosilanes to construct hydrazones with good functional group tolerance. The hydrazones can undergo nitrogen–nitrogen bond cleavage smoothly to deliver valuable amine derivatives. Additionally, asymmetric intermolecular hydroamination of unactivated aliphatic terminal alkenes using chiral *N*-imidazolinylphenyl 8-aminoquinoline (IPAQ) ligands has also been achieved to afford chiral amine derivatives with good enantioselectivities.

---

[1] Department of Chemistry, Zhejiang University, Hangzhou 310058, China. *email: luzhan@zju.edu.cn

Amine and its derivatives are significant in natural products and pharmaceutical chemistry (Fig. 1)[1–3], (https://www.pharmacy-tech-test.com/top-200-drugs.html). So development of novel methodologies for the synthesis of various amines and their derivatives is highly desirable, particularly from simple and readily available starting materials.

Metal-catalyzed hydrofunctionalization of readily available alkenes with nitrogen sources is one of the most efficient methods for the synthesis of nitrogen-containing molecules; however, to achieve the high regio- and enantioselectivities of this transformation is still a challenge (Fig. 2a)[4–19]. Among several activation strategies for alkene hydroamination, metal-catalyzed hydrogen atom transfer (HAT) reaction exhibits great Markovnikov selectivity and chemoselectivity (Fig. 2b)[20]. Okamoto and coworkers reported an early example of cobalt-catalyzed hydronitrosation of styrenes with nitric oxide using a hydroborane salt as a hydrogen

**Fig. 1 Amine-containing drugs.** Representative examples of pharmaceutical and biologically active compounds of amine derivatives.

**Fig. 2 Strategies for transition metal-catalyzed alkene hydroamination. a** Transition metal-catalyzed hydroamination of alkenes. **b** Metal-catalyzed radical hydroamination of alkenes. **c** Ligand-promoted metal-catalyzed alkene radical hydroamination via HAT. **d** Ligand-promoted metal-catalyzed asymmetric radical hydroamination of alkenes via HAT.

source to form oximes[21]. Since Mukaiyama and coworkers[22] reported the iron-catalyzed hydroamination of unactivated alkenes via HAT, using phenyl silane as a reductant and butyl nitrite as an aminating reagent, various aminating reagents, such as azo compounds[23-26], nitro compounds[27,28], diazo compounds[29], azides[24,30], and amides[31,32] have been explored by many research groups, which offers a great opportunity for retrosynthetic possibility of new transformations. Although simple 1,3-dicarbonyl metal complexes could promote the reactions, stoichiometric amount or high loading of these complexes used to be necessary[20,33]. Additionally, due to the generation of radical intermediates, asymmetric intermolecular hydroamination of alkenes via radical HAT process has not been explored. So the development of new types of ligands for efficient radical hydroamination of alkenes via a HAT process is still highly desirable.

It should be noted that the additional ligands used to promote the formation of metal hydride species that could undergo classic alkene insertion[34,35]. Due to the inhibition of generation of the alkyl radical, the reactivity and selectivity of transformation might decrease dramatically. Otherwise, the tetra- or more dentate ligand-based metal complexes could promote the generation of radical; however, no highly enantioselective examples have been reported so far due to the weak coordination of alkyl radical with metal complexes. So the discovery of suitable ligand scaffolds for metal-catalyzed radical hydroamination of alkenes is a challenge and also has great potential (Fig. 2c).

Continuing our pursuit of efficient earth-abundant transition metal catalysis via ligand design[36-45]; here, we report the use of unsymmetric *NNN* tridentate ligands to promote the cobalt-catalyzed radical hydroamination of alkenes via HAT (Fig. 2d). Meanwhile, asymmetric reaction of unactivated aliphatic terminal alkenes using chiral *N*-imidazolinylphenyl 8-aminoquinoline (IPAQ) ligand has also been achieved to afford chiral amine derivatives with good enantioselectivities.

## Results

**Reaction optimization.** The reaction of styrene **1a** with ethyl 2-diazo-2-phenylacetate **2a** in the presence of phenyl silane as a hydrogen donor in a solution of tetrahydrofura (THF) was selected as a model reaction. The reaction using Co(acac)₂ afforded the hydroamination product **3a** in a poor yield (entry 1, Table 1). The use of Co(OAc)₂ did not promote this transformation (entry 2). The known ligands for HAT, such as tpp, dmg, dppe, salen, and tridentate half salen **L1** have been tested; however, no reactivity or poor yields were observed (entry 3 and also see in Supplementary Table 1). *NN*-bidentate ligands such as 2,2′-bipyridine (bpy) and 1,10-phenanthroline (phen) could accelerate this reaction to give **3a** in 44 and 37% yields, respectively (entries 4 and 5). The reaction using *NNN* tridentate *N*-oxazolinylphenyl picolinamide (OPPA) ligands[42,46] **L2** and **L3** afforded **3a** in 88 and 92% yields, respectively (entries 6 and 7). Then, 8-aminoquinoline group was used as a directing group on the ligand instead of 2-picolinamide. The reaction using *N*-oxazolinylphenyl 8-aminoquinoline (OPAQ) **L4** as a ligand delivered **3a** in 96% yield (entry 8). So, the standard conditions were identified as 0.36 mmol of alkene **1**, 0.3 mmol of α-diazo ester compound **2**, 5 mol% of Co(OAc)₂, 6 mol% of **L4**, and 1.2 equiv. of PhSiH₃ in a solution of THF (0.25 M) at room temperature (r.t.) for 12 h.

**Substrate scope.** With the optimized conditions in hand, the substrate scope was shown in Table 2. A variety of styrenes bearing electron-donating or electron-withdrawing substituents at *para*-, *meta*-, or *ortho*-position on the phenyl ring underwent the reactions to afford the corresponding benzylamine derivatives (**3b–3t**) in 50–95% yields. For more broad synthetic interests, various functional groups, such as halide, ether, aniline, organoboronate, sulfide, nitrile, aldehyde, ester, and free alcohol were well tolerated. Meanwhile, 2-naphthyl (**1u**), pyridine derived alkenes (**1v** and **1w**) could be delivered to the corresponding

---

**Table 1 Optimizations for cobalt-catalyzed hydroamination of styrene.**

| Entry | [Co] | Ligand | Yield of **3a** (%)[b] |
|-------|------|--------|----------------------|
| 1 | Co(acac)₂ | – | 7 |
| 2 | Co(OAc)₂ | – | 0 |
| 3 | Co(OAc)₂ | **L1** | 0 |
| 4 | Co(OAc)₂ | **bpy** | 44 |
| 5 | Co(OAc)₂ | **phen** | 37 |
| 6 | Co(OAc)₂ | **L2** | 88 |
| 7 | Co(OAc)₂ | **L3** | 92 |
| 8 | Co(OAc)₂ | **L4** | 96 |

[a]The reaction was conducted using **1a** (0.36 mmol), **2a** (0.3 mmol), [Co] (5 mol %), ligand (6 mol%), PhSiH₃ (0.36 mmol) and THF (1.2 mL) under N₂ at r.t. for 12 h. [b]Determined by ¹H NMR using TMSPh as an internal standard.

[a]The reaction was conducted using **1a** (0.36 mmol), **2a** (0.3 mmol), [Co] (5 mol %), ligand (6 mol%), PhSiH₃ (0.36 mmol), and THF (1.2 mL) under N₂ at r.t. for 12 h
[b]Determined by 1H NMR using TMSPh as an internal standard

**Table 2 Reaction scope[a].**

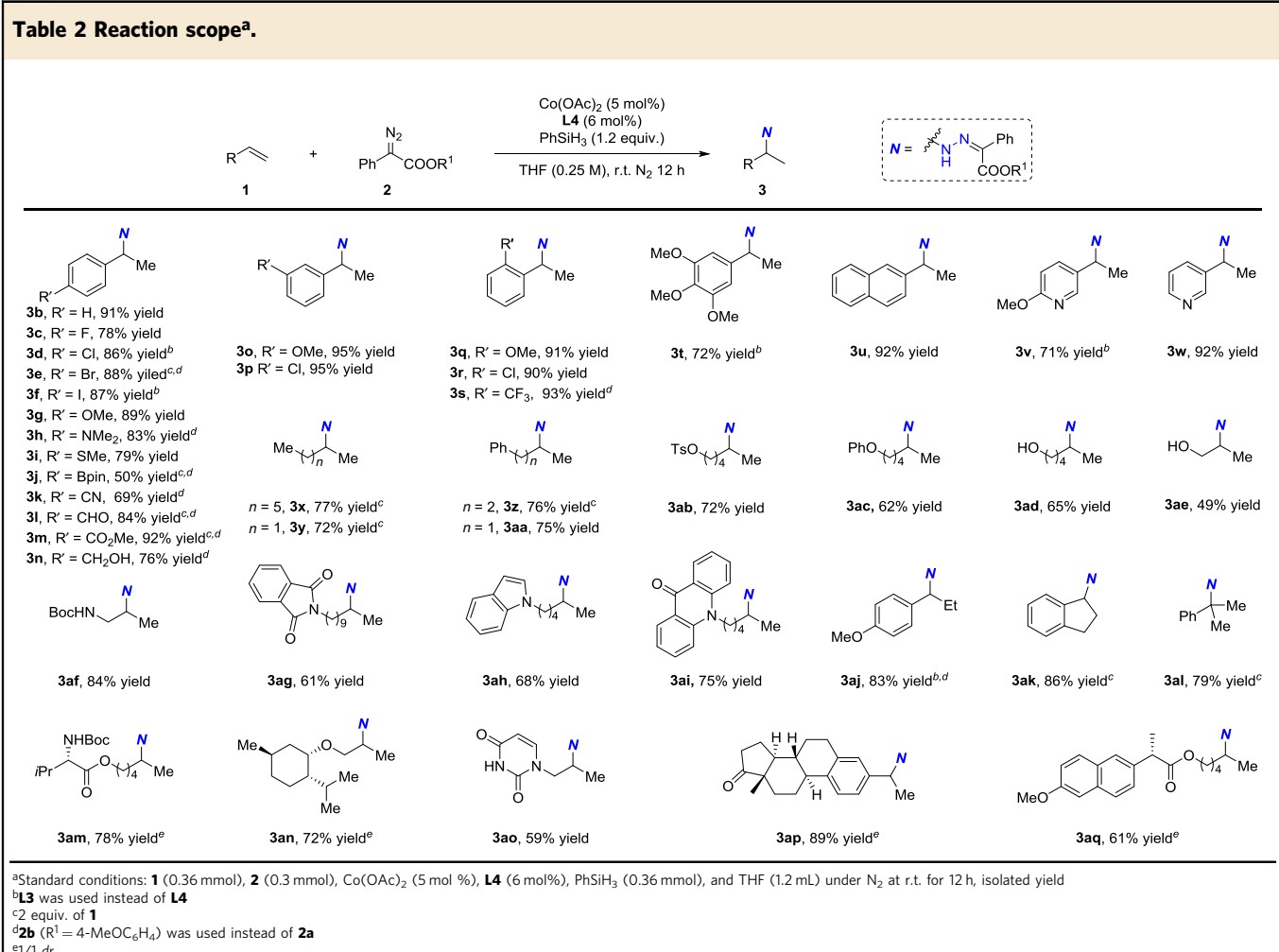

**3b**, R′ = H, 91% yield
**3c**, R′ = F, 78% yield
**3d**, R′ = Cl, 86% yield[b]
**3e**, R′ = Br, 88% yiled[c,d]
**3f**, R′ = I, 87% yield[b]
**3g**, R′ = OMe, 89% yield
**3h**, R′ = NMe₂, 83% yield[d]
**3i**, R′ = SMe, 79% yield
**3j**, R′ = Bpin, 50% yield[c,d]
**3k**, R′ = CN, 69% yield[d]
**3l**, R′ = CHO, 84% yield[c,d]
**3m**, R′ = CO₂Me, 92% yield[c,d]
**3n**, R′ = CH₂OH, 76% yield[d]

**3o**, R′ = OMe, 95% yield
**3p** R′ = Cl, 95% yield

**3q**, R′ = OMe, 91% yield
**3r**, R′ = Cl, 90% yield
**3s**, R′ = CF₃, 93% yield[d]

**3t**, 72% yield[b]

**3u**, 92% yield

**3v**, 71% yield[b]

**3w**, 92% yield

*n* = 5, **3x**, 77% yield[c]
*n* = 1, **3y**, 72% yield[c]

*n* = 2, **3z**, 76% yield[c]
*n* = 1, **3aa**, 75% yield

**3ab**, 72% yield

**3ac,** 62% yield

**3ad**, 65% yield

**3ae**, 49% yield

**3af**, 84% yield

**3ag**, 61% yield

**3ah**, 68% yield

**3ai**, 75% yield

**3aj**, 83% yield[b,d]

**3ak**, 86% yield[c]

**3al**, 79% yield[c]

**3am**, 78% yield[e]

**3an**, 72% yield[e]

**3ao**, 59% yield

**3ap**, 89% yield[e]

**3aq**, 61% yield[e]

[a]Standard conditions: **1** (0.36 mmol), **2** (0.3 mmol), Co(OAc)₂ (5 mol %), **L4** (6 mol%), PhSiH₃ (0.36 mmol), and THF (1.2 mL) under N₂ at r.t. for 12 h, isolated yield
[b]**L3** was used instead of **L4**
[c]2 equiv. of **1**
[d]**2b** (R¹ = 4-MeOC₆H₄) was used instead of **2a**
[e]1/1 *dr*

products in 71–92% yields. The reaction of alkyl-substituted terminal alkenes could undergo smoothly. Simple alkenes, such as 1-octene and 1-butene could be delivered to aliphatic amines in 77 and 72% yields, respectively. Alkenes, bearing phenyl, ether, tosyl, free alcohol, amine, amide, as well as nitrogen containing heterocycles, could be converted to the corresponding products in 49–84% yields. Notably, allylic alcohol and allylic amine derivatives could be reacted to deliver amino alcohol **3ae** and diamine **3af** in 49 and 84% yields, respectively. 1,1- and 1,2-disubstituted alkenes were suitable substrates and provided **3aj**–**3al** in 79–86% yields. Natural products, such as amino acid, menthol, uracil, and estrone bearing a terminal alkene moiety could be employed to deliver products (**3am**–**3ap**) in 59–89% yields, which also demonstrated that this method could be suitable for late-stage functionalization of complicated molecules. Terminal alkene bearing anti-inflammatory drug naproxen was converted to **3aq** in 61% yield. Configuration was confirmed by the X-ray diffraction of **3m**, and the other products were then assigned by analogy to **3m**.

**Further applications**. The gram-scale reaction could be smoothly performed to afford **3a** in 94% yield (Fig. 3a). The reaction of **3m** with 1,3-diketone afforded the pyrazole **4** in 86% yield (Fig. 3b). The three-step reactions containing alkene hydroamination, zinc-promoted reductive cleavage of N–N bond, and acyl protection of amine could be realized smoothly with once flash column chromatography purification process to deliver the corresponding

amides **6** and **7** in 53–75% yields (Fig. 3c). The pharmaceutical substances and biologically active amines **8**, such as appetite-suppressant drug clobenzorex, antianginal drug fendiline, and anti-hyperparathyroidism active NPS *R*568, could be efficiently synthesized in 32–45% yields from simple alkenes using a three-step synthetic protocol (Fig. 3c).

**Substrate scope of asymmetric hydroamination**. It should be noted that metal-catalyzed highly enantioselective hydroamination of unactivated aliphatic terminal alkenes is still an unsolved problem[47]. A primary study on asymmetric reaction using chiral unsymmetric *NNN*-tridentate ligand was explored. A chiral *N*-imidazolinylphenyl 8-aminoquinoline (IPAQ) ligand was designed and synthesized for asymmetric hydroamination of unactivated aliphatic terminal alkenes followed by cleavage of N–N bond and benzyl protection to afford chiral amine derivatives with up to 92.7 : 7.3 *er* (Table 3)[48]. The scope of substrate was quite broad. Various functional groups, such as halide, ether, and indole, could be tolerated. Alkene-bearing free alcohol moiety could underdo this transformation to afford the corresponding amide with a decreased yield. Chiral products with up to 98.6 : 1.4 *er* could be obtained after recrystallization. Particularly, the reaction of 1-butene afforded corresponding amide in 67% yield and 89.0 : 11.0 *er* (97.3 : 2.7 *er* after recrystallization). The chiral antihypertensive drug labetalol could be obtained from **14** via a known procedure[49].

**a** Gram-scale synthesis of **3a**

$$\mathbf{1a} \quad + \quad \mathbf{2a} \quad \xrightarrow{\text{Standard conditions}} \quad \mathbf{3a} \quad \begin{array}{l} \text{94\% yield} \\ \textbf{1.6498 g} \end{array}$$

**b** Synthesis of heterocycle

**c** Three steps synthesis of amines and its derivatives from alkenes

**6a,** R = 4-$t$BuC$_6$H$_4$, 55% yield
**6c,** R = 4-FC$_6$H$_4$, 70% yield
**6f,** R = 4-IC$_6$H$_4$, 55% yield
**6o,** R = 3-MeOC$_6$H$_4$, 75% yield
**7aa,** R = PhCH$_2$, 53% yield

**6ak,** 60% yield

**7al,** 59% yield

**8aa, Clobenzorex**
42% yield

**8b, (+/−)-Fendiline**
35% yield

**8o, (+/−)-NPS R568**
32% yield

**Fig. 3 Synthetic applications. a** Gram-scale synthesis. **b** Heterocycle synthesis. **c** Three steps synthesis of amines and its derivatives.

**Mechanistic studies**. Control experiments were conducted to illustrate the possible mechanism. The reaction in the presence of TEMPO did not occur, which indicated a possible radical pathway (Fig. 4a). The reaction of vinyl cyclopropane **28** afforded the ring-opening product **29** in 62% yield (Fig. 4b) via cleavage of the more substituted carbon–carbon bond[50], which supported the radical promoted ring-opening pathway[51]. A deuterium-labeled experimental reaction of indene using PhSiD$_3$ was conducted to afford **30** in 89% yield with 94% D and 1/1 *dr* (Fig. 4c) which demonstrated that the hydrogen came from hydrosilane and the carbon-centered radical formed as an intermediate.

Based on the experimental studies and previously reported literatures[24,27,52–56], a possible mechanism was shown in Fig. 5. The cobalt hydride species **A** obtained from the reaction of Co(OAc)$_2$ with ligand and silane could undergo a metal hydride HAT process to deliver the carbon radical intermediate and cobalt species **B**. Due to the possible redox non-innocent property of the ligand, the chemical valence of cobalt was not consistent. The cobalt species **B** might coordinate with diazo compound and undergo one electron oxidation with the carbon radical intermediate to afford the cobalt–carbon species **C**, which could undergo alkyl group migration from cobalt to nitrogen atom to generate cobalt oxide species **D**. The possibility that carbon radical directly attacked the cobalt coordinated diazo compound could not be ruled out. The cobalt species **D** could react with hydrosilane to regenerate the cobalt hydride species **A** and afford the vinyl silyl ether intermediate, which could undergo hydrolysis and isomerization to give the product. Further studies are

undergoing in our laboratory to gain an accurate understanding of the mechanism.

**Discussion**

In summary, we reported the use of unsymmetric *NNN*-tridentate OPAQ ligand to promote the cobalt-catalyzed radical hydroamination of alkenes via HAT. The protocol uses simple and commercially available alkenes to deliver the amination products with good functional group tolerance and high Markovnikov selectivity. The hydrazone compounds could undergo nitrogen–nitrogen bond cleavage smoothly to afford a series of biologically active molecules. In particularly, asymmetric reaction of unactivated aliphatic terminal alkenes using newly developed chiral IPAQ ligand has also been achieved to afford chiral hydroamination products with good enantioselectivity. Further studies on asymmetric hydrofunctionalization of simple alkenes are undergoing in our laboratory.

**Methods**

**Materials**. For the optimization of reaction conditions and control experiments of alkene 1a (Supplementary Table 1), and for the experimental procedures and analytic data of compounds synthesized (Supplementary Methods). For nuclear magnetic resonance (NMR) spectra of compounds in this manuscript (Supplementary Fig. 1–166). For high-performance liquid chromatography (HPLC) spectra of compounds in this manuscript (Supplementary Fig. 167–185).

**General procedure A for hydroamination of alkenes**. A 25 mL Schlenk flask equipped with a magnetic stirrer and a flanging rubber plug was dried with flame under vacuum. When cooled to ambient temperature, it was vacuumed and flushed

**Table 3 Reaction scope of asymmetric hydroamination[a].**

9, n = 1, 67% yield, 89.0 : 11.0 er (97.3 : 2.7 er after recrystal.)
10, n = 3, 60% yield, 90.5 : 9.5 er
11, n = 5, 63% yield, 92.7 : 7.3 er (97.6 : 2.4 er after recrystal.)
12, n = 7, 78% yield, 92.2 : 7.8 er
13, n = 9, 59% yield, 92.3 : 7.7 er

14, n = 2, 48% yield, 90.3 : 9.7 er (98.6 : 1.4 er after recrystal.)
15, n = 4, 74% yield, 92.6 : 7.4 er

16, 54% yield, 91.6 : 8.4 er

17, 64% yield, 91.8 : 8.2 er

18, 71% yield, 91.5 : 8.5 er

19, 61% yield, 91.6 : 8.4 er

20, 62% yield, 91.9 : 8.1 er

21, 74% yield, 90.7 : 9.3 er

22, 60% yield, 91.7 : 8.3 er

23, 73% yield, 91.4 : 8.6 er

24, 73% yield, 88.5 : 11.5 er

25, 38% yield, 89.6 : 10.4 er

26, 64% yield, 91.2 : 8.8 er

27, 82% yield, 92.0 : 8.0 er

14 (98.6 : 1.4 er after recrystal.) →(ref. 50) (S,R) or (R,R)-labetalol

[a]Standard conditions: (1) **1** (0.6 mmol), **3c** (0.3 mmol), Co(OAc)$_2$ (5 mol%), IPAQ (6 mol%), PhSiH$_3$ (0.6 mmol), 2-ethoxyethanol (3 equiv.), and ethyl acetate (1.2 mL) under N$_2$ at –10 °C for 24 h; (2) zinc dust (25 equiv.) and AcOH–THF–H$_2$O (3/1/1, v/v/v, 3 mL) under air at 60 °C for 3 h; and (3) BzCl (1.5 equiv.), Et$_3$N (2 equiv.), and THF (3 mL) under air at r.t. for 2 h; isolated yield, er was determined by HPLC

**Fig. 4 Mechanistic studies. a** Radical trapping experiment. **b** Radical clock experiment. **c** Deuterium-labeling experiment. TEMPO, 2,2,6,6-tetramethylpiperidine-1-oxyl.

**Fig. 5 Proposed mechanism.** Cobalt-induced HAT generates carbon radicals that rebind to cobalt followed by attack on diazo compound and formation of C–N bond.

with N$_2$ and repeated for three times. To the flask, Co(OAc)$_2$ (0.015 mmol), **L3** or **L4** (0.018 mol), and THF (1.2 mL) were added. The flask was degassed and stirred for 30 min at r.t. Then, PhSiH$_3$ (0.36 mol), diazo compound (0.3 mmol), and alkene (0.36 mmol) were added in sequence. After 12 h, the reaction was quenched with 10 ml of petroleum ether (PE) and the mixture was filtered through a pad of silica gel and washed with PE/EtOAc (5/1, 50 mL). The combined filtrates were concentrated and purified by flash column chromatography using PE/EtOAc as the eluent to afford the corresponding product.

**General procedure B for asymmetric hydroamination of alkenes**. A 25 mL Schlenk flask equipped with a magnetic stirrer and a flanging rubber plug was dried

with flame under vacuum. When cooled to ambient temperature, it was vacuumed and flushed with N$_2$ and repeated for three times. To the flask, Co(OAc)$_2$ (0.015 mmol), IPAQ (0.018 mol), ethyl acetate (1.2 mL), and 2-ethylethanol (100 μL, 0.93 g/mL, 0.9 mmol) were added. The flask was degassed and cooled down to −10 °C and stirred for 30 min. Then, PhSiH$_3$ (0.36 mol), diazo compound (0.3 mmol), and alkene (0.36 mmol) were added in sequence. After 24 h, the reaction was warmed up to r.t. and quenched with 10 ml of PE. The mixture was filtered through a pad of silica gel and washed with PE/EtOAc (5/1, 50 mL). The combined filtrates were concentrated to afford a yellow oil that was used for the next step without further purification.

**Cleavage of *N–N* bond**. To the above suspension, AcOH–THF–H$_2$O (3:1:1 v/v/v, 3 mL) was added, followed by addition of activated Zn powder (0.5 g, 7.5 mmol) in several portions at r.t. After that, the mixed solution was warmed up to 60 °C and stirred until completion monitored by thin-layer chromatography (usually 3 h). Then, the reaction mixture was cooled down to r.t. and quenched with water (20 mL). The reaction mixture was basified with a solution of NaOH (6 N) until the solution turned clear (pH > 10) and then extracted with Et$_2$O (20 mL × 4). The combined organic layers were dried over anhydrous Na$_2$SO$_4$, filtered, concentrated to give a yellow oil that was used for the next step without further purification.

**Protection of free amines with Bz group**. To the above oil, 3 mL (0.1 M) of THF, 55 μL (1.211 g/mL, 0.45 mmol) of BzCl, and 84 μL (0.728 g/mL, 0.6 mmol) of Et$_3$N were added, followed by 2 h stirring. The mixture was quenched with water (20 mL) and then extracted with Et$_2$O (20 mL × 4). The combined organic layers were dried over anhydrous Na$_2$SO$_4$, filtered, concentrated, and purified by flash column chromatography using PE/EtOAc as the eluent to give the corresponding amide.

## Data availability

The authors declare that the data supporting the findings of this study are available within the paper and its Supplementary Information file. The X-ray crystallographic coordinates for structures for **3m** has been deposited at the Cambridge Crystallographic Data Centre (CCDC) under deposition numbers CCDC 194446. The data can be obtained free of charge from the Cambridge Crystallographic Data Centre via http://www.ccdc.cam.ac.uk/data_request/cif. The experimental procedures and characterization of all new compounds are provided in Supplementary Information.

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

## Acknowledgements

Financial support was provided by NSFC (21922107 and 21772171), Zhejiang Provincial Natural Science Foundation of China (LR19B020001), Center of Chemistry for Frontier Technologies, and the Fundamental Research Funds for the Central Universities (2019QNA3008).

## Author contributions

Z. L. designed the experiments. X. S., J. C. and Y. S. performed the experiments. Z. L. and X. C. designed the racemic ligands. Z. L. and Z. C. designed the chiral ligands. Z. L. and X. S. prepared this manuscript. X. S., J. C., Y. S. and Z. L. prepared the supplementary information.

## Competing interests

The authors declare no competing interests.
