## [Peer Review File · Nature Communications]

Reviewers'

comments:

Reviewer #1 (Remarks to the Author):

Lu and co-workers report Markovnikov-type selective radical hydroamination of alkenes with diazo compounds by means of hydrogen atom transfer. The tridentate ligand effect and broad scope is quite intriguing. The authors synthesized the scalemic products using chiral ligand. The mechanism of this transformation is also quite intriguing. Unfortunately, the scope is too narrow (only two examples). It should be noted that Pronin recently reported the enantioselective hydroalkoxylation of tertiary allylic alcohol by hydrogen atom transfer -initiated process (JACS, 17532, 2019). A similar transformation using diazo compound has been reported Cui and co-worker, as the authors cited in ref. 30. The authors should correct the reference 19 in Scheme 2 (maybe 49). Overall, this work does not meet requirements of novelty to accept publication in Nature communication.

Reviewer #2 (Remarks to the Author):

In this submission Lu and coworkers have developed a highly efficient hydroamination reaction of terminal olefins and indenenes with diazo compounds, with the product being hydrazone derivatives, which are shown to be readily converted to useful structures as shown in Table 2c. The authors have demonstrated that the choice of the ligand (L2-4, which were developed by the same team) is crucial for the desired transformation. They have thoroughly investigated the substrate scope for this new reaction and proposed that reaction likely proceeds through a radical (HAT) mechanism. Most importantly, for the first time such alkyl radical interception event (by diazo) is achieved in high stereoselectivity (up to 85% ee).

I am enthusiastic about this paper and especially excited about the enantioselectivity demonstrated. Catalyst controlled stereoselectivity has long been elusive in first-row transition metal catalyzed HAT radical hydrofunctionalizations and this paper represent perhaps one of the first intermolecular examples with Co in high ee. This particular ligand class has rarely been considered in the context of HAT catalysis and the original work shown here will significantly broaden the horizon of catalytic HAT reactions. It would be very interesting to know how this catalyst system is so efficient to induce an inner-sphere functionalization process of an alkyl radical species, and I am looking forward to see more work on this from this team. Overall this is a very nice piece of work with high originality and likely will be impactful to the synthetic organic community. Therefore I strongly support publication as it is.

A few minor points:

- 1) Page 1, right column bottom, "it should be noted that..... classic alkene insertion", references are needed.
- 2) Scheme 3, the charges in C ("+" should be at the inner "N") and D need to be corrected.

Reviewer #3 (Remarks to the Author):

The nitrogen-containing compounds are very important and useful in many research areas or industry. Although metal-catalyzed hydroamination of alkene has been reported via HAT, the lack of efficient ligand did limit this type of transformation. Lu and co-workers reported an interesting cobalt-catalyzed radical-type hydroamination reaction of alkenes to access the nitrogen-containing compounds with excellent regioselectivity and functional group tolerance. In this manuscript, the novel unsymmetric NNN tridentate ligand was designed to efficiently promote the radical hydroamination of alkenes. The scope of substrates is quite broad. The gram-scale reaction could be conducted smoothly. Due to the ease of further derivatization, various amides and bioactive molecules could be obtained efficiently. Additionally, the first intermolecular asymmetric radical hydroamination of unactivated simple alkenes has been illustrated by using newly developed IPAQ ligand. The enantioselectivity is up to 92.5 :7.5 er which is the best result in intermolecular asymmetric hydroamination of aliphatic simple alkenes. All these results demonstrated the potential utility of this protocol. It may also open a new avenue for the highly enantioselective hydroamination of unactivated simple alkenes. The SI was well prepared. It should be noted that compared to the author's previous works on hydroboration and hydrosilylation of alkenes, this work may be a new direction. So this reviewer suggested that this manuscript was suitable for Nature Communications.

Comments:

1. The recent examples of metal-catalyzed relay asymmetric radical reactions could be cited, such as Acc. Chem. Res. 2019, DOI:10.1021/acs.accounts.9b00381; Nat. Chem. 2019, 11, 1158.
2. To prove the utility of asymmetric transformation, more examples on asymmetric hydroamination

could be presented.

3. In the scheme 3, the Co(III) hydride species might form to undergo HAT. The chemical valence on cobalt should be mentioned in the catalytic cycle. If the ligand were non-innocent, the chemical valence "n" or "n+1" could be used for cobalt.

4. On left column in page 3, the word "complex" in "be suitable for late-stage functionalization of complex molecules" should be instead by "complicated".

5. On left column in page 3, the word "alcohols" in "Alkenes, bearing phenyl, ether, tosyl, free alcohols, amine," should be "alcohol".

Manuscript ID: NCOMMS-19-36006

Title: "Ligand Promoted Cobalt-Catalyzed Radical Hydroamination of Alkenes"

Author(s): Xuzhong Shen, Xu Chen, Jieping Chen, Yufeng Sun, Zhaoyang Cheng,
Zhan Lu

Reviewer #1 (Remarks to the Author):

Comment 1-1

The authors synthesized the scalemic products using chiral ligand. The mechanism of this transformation is also quite intriguing. Unfortunately, the scope is too narrow (only two examples).

Answer to Comment 1-1:

Thanks for your comment. 19 examples have been presented in Table 3, described as follow: "The scope of substrate was quite broad as similar as that of racemic reactions. Various functional groups, such as halide, ether, free alcohol and indole, could be tolerated."

Comment 1-2

It should be noted that Pronin recently reported the enantioselective hydroalkoxylation of tertiary allylic alcohol by hydrogen atom transfer -initiated process (JACS, 17532, 2019).

Answer to Comment 1-2:

Thanks for your comment. Pronin's work was quite intriguing for intramolecular asymmetric transformation via HAT process. And Pronin's work also suggested the possibility of HAT mechanism, thus, it has been cited as reference 56.

Comment 1-3

The authors should correct the reference 19 in Scheme 2 (maybe 49).

Answer to Comment 1-3:

Thanks for your comment. The reference in Scheme 2 (now Table 3) has been corrected in the revised manuscript.

Reviewer #2 (Remarks to the Author):

Comment 2-1

Page 1, right column bottom, "it should be noted that..... classic alkene insertion", references are needed.

Answer to comment 2-1:

Thanks for your advice. Selected reviews on hydrometallation of alkenes have been cited (reference 35 and 36) in the revised manuscript.

Comment 2-2

Scheme 3, the charges in C ("+" should be at the inner "N") and D need to be corrected.

Answer to comment 2:

Thanks for your comment. The charges in C and D in Scheme 3 (now Scheme 2) have been corrected in the revised manuscript.

Reviewer #3 (Remarks to the Author):

Comment 3-1

The recent examples of metal-catalyzed relay asymmetric radical reactions could be cited, such as Acc. Chem. Res. 2019, DOI:10.1021/acs.accounts.9b00381; Nat. Chem. 2019, 11, 1158.

Answer to comment 2:

Thanks for your advice. Recent reviews and examples of metal-catalyzed relay asymmetric radical reactions have been cited in the revised manuscript, as reference 57 and 58.

Comment 3-2

To prove the utility of asymmetric transformation, more examples on asymmetric hydroamination could be presented.

Answer to the comments:

Thanks for your advice. Substrate scope on asymmetric hydroamination of alkenes has been studied and shown in Table 3 in the revised manuscript. 19 examples have been added. “The scope of substrate was quite broad as similar as that of racemic reactions. Various functional groups, such as halide, ether, free alcohol and indole, could be tolerated.”

Comment 3-3

In the scheme 3, the Co(III) hydride species might form to undergo HAT. The chemical valence on cobalt should be mentioned in the catalytic cycle. If the ligand were non-innocent, the chemical valence “n” or “n+1” could be used for cobalt.

Answer to the comments:

Thanks for your advice. Due to the possible redox non-innocent property of the ligand, the exact chemical valence has not been confirmed yet. For example, the formal chemical valence of cobalt was changed from “n” in complex A to “n-1” in complex B, however, the exact chemical valence of cobalt in complex B may be not “n-1”. So the chemical valence was not described.

Comment 3-4

On left column in page 3, the word “complex” in “be suitable for late-stage functionalization of complex molecules” should be instead by “complicated” .

Answer to the comments:

Thanks for your advice. The word “complex” has been changed to “complicated”.

Comment 3-5

On left column in page 3, the word “alcohols” in “Alkenes, bearing phenyl, ether, tosyl, free alcohols, amine,” should be “alcohol”.

Answer to the comments:

Thanks for your advice. The word “alcohols” has been changed to “alcohol”.

The references on amine and its derivatives have been modified. The original ref. 1 was not suitable. So, the ref. 1 has been instead by “*Alkaloids: A Treasury of Poisons*”

and Medicines; Funayama, S.; Cordell, G. A. Eds.; Waltham, MA (2014).”

The original references(2-4) have been moved to follow the first sentence.

REVIEWERS'

COMMENTS:

Reviewer #1 (Remarks to the Author):

The author nicely explained the enantioselectivity of mechanism, and presented 19 asymmetric examples in the revised MS. Unfortunately, the ee% are not satisfactory. In my opinion, the enantioselectivity to be accepted in a high-impact journal should be more than 90%ee (one or two examples is enough). Otherwise, the author should disclose the story of optimization. Also, I would recommend the authors reconsider inclusion of the statement "the scope of substrate was quite broad as similar as that of racemic reactions". For examples, Table 3 does not cover many products in Table 2. I understand that any methods contain each deficiency in scope. At least, the author should clarify the limitation of this method in the Table 3 and sentence (include negative results).
-Reference 50: Correct the compound number. 7z is not shown in table 3.

Reviewer #2 (Remarks to the Author):

The authors have nicely addressed the review comments. I recommend publication as it is.

Reviewer #3 (Remarks to the Author):

I completely agree to publish it in Nature Communications after revision by authors

Journal: Nature Commutations

Manuscript ID: NCOMMS-19-36006A

Title: "Ligand Promoted Cobalt-Catalyzed Radical Hydroamination of Alkenes"

Author(s): Xuzhong Shen, Xu Chen, Jieping Chen, Yufeng Sun, Zhaoyang Cheng, Zhan Lu

Reviewer #1 (Remarks to the Author):

Comment 1-1: The author nicely explained the enantioselectivity of mechanism, and presented 19 asymmetric examples in the revised MS. Unfortunately, the ee% are not unsatisfactory. In my opinion, the enantioselectivity to be accepted in a high-impact journal should be more than 90%ee (one or two examples is enough).

Answer to Comment 1-1: Thanks for your comment. So far, we do our best to access chiral aliphatic amine derivatives with up to 92.6 : 7.4 *er*. To the best of our knowledge, it is the best *er* achieved in metal-catalyzed hydroamination of unactivated aliphatic terminal alkenes to date. Additionally, high enantioselective chiral amine derivatives with up to 98.6 : 1.4 *er* could be obtained via simple recrystallization.

Comment 1-2: Otherwise, the author should disclose the story of optimization.

Answer to Comment 1-2: Thanks for your comment. The primary optimization is disclosed as follow:

Entry	R	Solvent	additive	Temp. (°C)	Yield of 3 (%) ^a	Er	Note
1	Et	THF	w/o	0	47	88.7 : 11.3	SXZ5151B
2	Ph	THF	w/o	0	67	85.6 : 14.4	SXZ6071A
3	t Bu	THF	w/o	0	45	90.5 : 9.5	SXZ6016A
4	t Bu	DMF	w/o	0	16	83.6 : 16.4	SXZ7049A
5	t Bu	MeCN	w/o	0	22	89.7 : 10.3	SXZ7049B
6	t Bu	DCE	w/o	0	n.d. ^b	n.d. ^b	SXZ7049C
7	t Bu	EA	w/o	0	48	92.1 : 7.9	SXZ6156A
8 ^c	t Bu	EA	3 equiv. 2-Ethoxyethanol	-10	78	90.4 : 9.6	SXZ9015

^a Yields were determined by ¹HNMR using TMSPh as an internal standard. ^b not detected. ^c24 h.

Comment 1-3: Also, I would recommend the authors reconsider inclusion of the statement “the scope of substrate was quite broad as similar as that of racemic reactions”.

Answer to Comment 1-3: Thanks for your advices. After consideration, the text “the scope of substrate was quite broad as similar as that of racemic reactions” was changed into “the scope of substrate was quite broad”.

Comment 1-4: For examples, Table 3 does not cover many products in Table 2. I understand that any methods contain each deficiency in scope. At least, the author should clarify the limitation of this method in the Table 3 and sentence (include negative results).

Answer to Comment 1-4: Thanks for your advices. Due to time limitation, we do not cover all products in Table 2. Alkene bearing an indole moiety could undergo this transformation to afford the corresponding amide with a slight lower *er*. Alkenes bearing free alcohol gives corresponding product in a lower yield. These results have been added in the table 2.

Comment 1-5: Reference 50: Correct the compound number. 7z is not shown in table 3.

Answer to Comment 1-5: Thanks for your advices. The text “The absolute configuration was confirmed by comparing with the known chiral compound 7z, see:” from reference 50 is removed.

Reviewer #2:

The authors have nicely addressed the review comments. I recommend publication as it is.

Answer: It is appreciated.

Reviewer #3:

I completely agree to publish it in Nature Communications after revision
by authors

Answer: It is appreciated.